# Comparative Analysis of Human Body Temperatures Measured with Noncontact and Contact Thermometers

**DOI:** 10.3390/healthcare10020331

**Published:** 2022-02-09

**Authors:** Patrycja Dolibog, Barbara Pietrzyk, Klaudia Kierszniok, Krzysztof Pawlicki

**Affiliations:** Department of Medical Biophysics, Medical University of Silesia, 40-055 Katowice, Poland; pdolibog@sum.edu.pl (P.D.); klaudia.kierszniok@sum.edu.pl (K.K.); pawlicki@sum.edu.pl (K.P.)

**Keywords:** body temperature, thermometer types, COVID-19 screening

## Abstract

Body temperature measurement is one of the basic methods in clinical diagnosis. The problems of thermometry—interpretation of the accuracy and repeatability of various types of thermometers—are still being discussed, especially during the current pandemic in connection with the SARS-CoV-2 virus responsible for causing the COVID-19 disease. The aim of the study was to compare surface temperatures of the human body measured by various techniques, in particular a noncontact thermometer (infrared) and contact thermometers (mercury, mercury-free, electronic). The study included 102 randomly selected healthy women and men (age 18–79 years). The Bland–Altman method was used to estimate the 95% reproducibility coefficient, i.e., to assess the degree of conformity between different attempts. Temperatures measured with contact thermometers in the armpit are higher than temperatures measured without contact at the frontal area of the head. The methods used to measure with contact thermometers and a noncontact infrared thermometer statistically showed high measurement reliability. In order to correctly interpret the result of measuring human body temperature, it is necessary to indicate the place of measurement and the type of thermometer used.

## 1. Introduction

Temperature is one of the most frequently measured physical quantities in medical practice. Temperature measurement provides information about the internal energy of an object, and its determination and control are of significant diagnostic importance. From a biophysical point of view, temperature measurement determines changes in physical quantities that occur in a thermodynamic system. It consists of determining the difference between the temperature of the measured medium and the temperature conventionally considered to be 0 on the temperature scale, which in turn, assigns appropriate numerical values to a given temperature range [1]. The most commonly used temperature scales are: Celsius, Fahrenheit, Kelvin and Rankine [2]. Human skin temperature is the result of a dynamic balance between the heat released in metabolic transformations and delivered to the skin layer (by conduction and convection), the heat released to the environment (by radiation, convection, and evaporation), and the heat extracted from the environment. To determine the surface temperature of the human body, we most often use medical thermometers. The way in which heat is transferred between the thermometer and the body is the basic criterion for the division of thermometric instruments. We distinguish between contact and noncontact devices. Noncontact thermometers measure the infrared radiation emitted by an object and convert the detected energy into a temperature value. They are correctly called radiation thermometers or pyrometers [3]. Contact (touch) thermometers, due to the properties of the thermometric body used in the device, can be divided into liquid (mercury-withdrawn, in Poland and other European Union countries, from production and sales in 2009) or alcohol, bimetallic, gas, electric and magnetic. Their operation is based, among others, on changes in volumetric or linear expansion, resistivity, thermoelectric force, or the difference of potentials at the contacts of various materials. The heat exchange that takes place in this case between the measured object and the temperature sensor is primarily based on conduction. Contact thermometers allow you to measure body temperature in the oral cavity, in the armpit or in the rectum [4]. Another division criterion, which depends on the technique used to convert the measured values into temperature values, is the division into electronic and nonelectronic thermometers.

The principle of operation of nonelectronic thermometers is based on the use of the phenomenon of liquid volume expansion under the influence of supplied heat. On the other hand, practically all electronic devices used for temperature measurement require the use of electronic systems to process the measured voltage or current signal, enabling its linearization, elimination of interferences, and conversion of the measured quantity into temperature values and amplification. Professional electronic thermometers allow you to perform a single measurement or monitor the temperature for a specified period. Moreover, the use of disposable thermometer probe covers reduces the risk of transmission of pathogens [5]. Traditional mercury thermometers are highly reliable, but due to the harmfulness of mercury vapor, as mentioned, they are withdrawn from use in most European Union countries. Reading from liquid thermometers is not as fast as with electronic thermometers and takes up to 5 min [6]. The sensitivity of the thermometer is the ratio of the mercury displacement per unit temperature. We can increase it by reducing the diameter of the capillary and increasing the volume of the liquid reservoir, which increases the thermal inertia of the thermometer.

For typical clinical thermometers, the accuracy is about 0.1 °C. Electronic thermometers are equipped with a digital display, which greatly facilitates and improves the reading. Their disadvantage is the fact that, like traditional thermometers, they are contact devices, which require the implementation of disinfecting agents [7]. Only pyrometers are devices that provide quick and noncontact temperature measurements (skin layers), which makes them the most frequently chosen measuring device for Covid-19 prophylaxis. The measurement result is influenced by the emissivity and reflection coefficient of the emitting skin surface, the transmission coefficient of the medium between the sensor and the target, the average radiation temperature of the measuring environment as well as the sensor’s distance from the tested target. In the case of this type of thermometer, the measurement error is approximately 0.2–0.4 °C [8]. Depending on the type of thermometer, measurements are most often taken in the armpit, mouth, rectum, ear, or forehead. The normal temperature value in the armpit is usually 36.6 °C, in the mouth 36.9 °C, in the ear—the correct temperature is 37.1 °C. Ear measurement values are assumed to be of similar accuracy compared to the rectal method most commonly used in infants, the correct value is usually 37.1 °C.

The rectal temperature test method is the most accurate of all measurements, while noncontact forehead thermometers are considered the least accurate, and measurements carried out using them should be confirmed by other methods. The minimum value of the standard error for the above methods is 0.1 °C [9].

The result of temperature measurement with contact thermometers is less dependent on external factors compared to the results of temperature measurements made with pyrometers. No significant influence of external factors (ambient temperature, air humidity, exposure sunlight) on the results of the temperature measurements was observed [10].

Body temperature is a physiological response controlled by a thermoregulation set point programmed by the hypothalamic thermoreceptors (set point) and conditioned by the body’s precise thermoregulatory systems. The change in body temperature, in relation to the normal temperature, is observed during the ongoing disease processes in the body. It is the result of a change in the physical and chemical properties of tissues, such as thermal conductivity, pH, or the breakdown of electrolytes. An imbalance in the calcium-sodium balance in the hypothalamus causes a change in body temperature. Increasing the calcium ions in the hypothalamus causes the biological thermostat to shift to a lower temperature and the body temperature to decrease. Increasing the concentration of sodium ions causes the biological thermostat to switch to a higher temperature and, consequently, to an increase in body temperature. This may lead to a fever [11]. Physiological temperature values depend on the place of testing and the method of measurement as well as the type of thermometer.

The first step in the fight against a pandemic is the early diagnosis of patients with infections, then their isolation and implementation of treatment procedures.

Nevertheless, early detection of these patients can be difficult, because 50% of COVID-19 infected patients are so-called “asymptomatic” and do not have a fever until the disease develops [12,13,14]. In turn, in the remaining 50% of people, a fever appeared as the first symptom of infection [15]. Symptoms are not sufficient to diagnose COVID-19, medical laboratory tests are required. However, from the perspective of screening in a large population, fever is one of the main symptoms of COVID-19 infection, and a method and appropriate tools for measuring temperature are necessary for the correct assessment of disease progression.

Wunderlich, already at the end of the nineteenth century, gave 37.0 °C as the average normal value of a healthy person’s body temperature within the normal range 36.2–37.5 °C. He obtained the results by studying the surface temperature of about 25,000 people using a mercury thermometer measured under the armpit [16]. He determined the change in the value of the surface temperature of the body depending on the time of day, which reaches its minimum between 2 a.m. and 8 a.m., and the maximum between 4 p.m. and 9 p.m. [17].

In turn, in a study of 45 people, Zhen Chen and others demonstrated the clinically significant advantage of noncontact infrared thermometers over mercury armpit thermometers and infrared eardrum thermometers in body temperature measurement [18]. Wenxi Chen’s review confirms that, despite the different types of thermometers and differences in measurement values between them, depending on the location of the test, the type and sensitivity of the equipment, body temperature measurement remains the basic diagnostic tool in the assessment of the symptoms of an inflammatory disease. The research therefore confirms the validity of temperature measurement as a basic screening test for the detection of COVID-19. W. Chen draws attention to the importance of real-time thermometer diagnosed based on the statistical forecasting of a large population, rightly suggesting personalization and an individual diagnostic approach to long-term application as part of the adaptive modeling method [19].

Factors that cause an imbalance of calcium and sodium in the hypothalamus include pyrogens, including viruses. For this reason, in case of suspected coronavirus infection, the preliminary diagnostic screening method is temperature measurement. Most often, noncontact thermometers are used, commonly referred to as non-contact. This is a relatively sensitive method, although the measurement conditions must be strictly defined in the case of these thermometers. The measurement should be made on clean, dry skin. Their advantage is noninvasiveness, as well as hygiene, enabling multiple temperature measurements without disinfection of the device each time [20]. Infection with the SARS-CoV-2 virus is responsible for causing the COVID-19 disease (coronavirus disease 2019). In some people, an asymptomatic course of infection is observed, while in others, symptoms with varying severity appear. Among them, a characteristic first symptom of the disease is an increase in body temperature. Fever can vary in severity from slightly elevated to 40 degrees Celsius, followed by typical COVID-19 symptoms such as shortness of breath, cough, muscle aches and chills, headaches, loss of taste and olfaction, nausea, and vomiting. The most common complication of the disease is pneumonia, which progresses relatively quickly if untreated in case of infection with the SARS-CoV-2 virus, leading to acute respiratory failure [21]. The occurrence of fever is individually variable and depends on factors such as age, physical activity, stress, fatigue, metabolic activity, and immunity of the body. Fever can be characterized by a different course and level of temperature increase, from mild to acute, short, or long duration—from one day to the persistence of elevated temperature for several weeks. During coronavirus infection, it is an important diagnostic criterion. However, it belongs to nonspecific symptoms—many disease states are manifested by increased body temperature. Despite differences depending on the above factors and the age of the patient, body temperature measurement during the pandemic has become and continues to be a common and fast primary diagnostic screening method for suspected COVID-19 [22]. It is worth remembering, that temperature screening can be highly error prone resulting from a lack of skills and ignorance about measurement techniques. Identifying individuals with raised temperature in the context of infection is important to inhibit the spread of virus and assess the rate of transmission in the population. It is also important for taking further actions regarding both the decision to isolate people infected with coronavirus and further diagnostics for COVID-19 [23].

The aim was to compare the surface temperatures of the human body measured by various techniques, in particular a noncontact thermometer (infrared) and contact thermometers (mercury, mercury-free, electronic). A novelty in our research was that the temperature was measured under controlled conditions using several methods (thermometers) in the same healthy people.

## 2. Materials and Methods

### 2.1. Materials

One hundred and two randomly selected adults, healthy women and men aged 18–79 years, were enrolled in the survey.

The authors created a questionnaire for this study, based on interview with the patient during a routine visit, after receiving consent to participate in this survey. The study was performed anonymously, without collection of patients’ personal data. The study was conducted as a survey that did not fulfill the medical experiment criteria and therefore did not require Bioethical Committee approval.

The study included 50 women aged 18–50 years (mean age 32.4 ± 9.7 years) and 52 men aged 20 to 79 years (mean age 45.7 ± 12.6 years) (Table 1). In all subjects, the surface temperature was recorded with four types of thermometers: T1—mercury contact thermometer, T2—mercury-free contact thermometer, T3—electronic (resistive) contact thermometer, T4—noncontact infrared thermometer. Before measuring the temperature with traditional glass thermometers (mercury T1 contact and mercury-free T2 contact), the thermometer had to be shaken so that the liquid bar indicating the temperature value was set below 35.5 °C. The thermometer was placed in the armpit, in such a way that its tip closely adhered to the skin, and then the arm was pressed. During the measurement, which lasted 5 min, the patient did not move and remained in comfort. After the measurement was completed, the temperature was read. Then the thermometer was disinfected. Temperature measurement with the T3 electronic contact thermometer was also made in the armpit. After pressing the “on” button, a message appeared on the thermometer display allowing the measurement to begin. The measurement lasted until the moment of sound signaling, usually no longer than 1 min. Then the temperature was read from the display and the thermometer was disinfected. The noncontact infrared thermometer T4 measured the temperature at the forehead. After turning on the thermometer, it was brought to a distance of 2–2.5 cm from the forehead (perpendicular to the forehead). Then the measurement was started by holding the appropriate button. The measurement result appeared on the display and additionally the color of the button illumination changed, indicating if the temperature was normal (green) or not correct (red). After reading the temperature, the thermometer was disinfected. All measurements were always made at the same hours from 3–4 p.m. in the same room, while maintaining a constant ambient temperature of 23 °C. Temperature measurements were made by qualified medical personnel with absolute observance of sanitary and epidemiological restrictions. To optimize the measurements and accommodation to the temperature in the measuring room, the subjects remained at rest in a sitting position for 20 min before proceeding with the first measurement and 5 min between subsequent measurements. Each temperature measurement (with different thermometers T1, T2, T3 and T4) was performed once by one person.

### 2.2. Data Analysis

Statistical analysis of the examined parameters was presented for descriptive statistics and tests: ANOVA, *t* student, post hoc RIR Turkey and coefficient of intraclass variability. The values of tests and coefficients at the *p* level < 0.05 were considered statistically significant in all tests carried out.

## 3. Results

One hundred and two adults agreed to participate in the measurements. All temperatures were normal, correct, and did not indicate fever or hypothermia

The average temperature value measured with a mercury contact thermometer was 36.2 °C, standard deviation 0.5 °C, and the median was equal to 36.2 °C. The average temperature value measured with a mercury-free contact thermometer was 36.3 °C, the median was equal to 36.4 °C, and standard deviation 0.5 °C. The average value of the temperature measured with an electronic contact thermometer was 36.2 °C, the median was equal to 36.3 °C and standard deviation 0.5 °C. In turn, the average temperature value measured with an infrared noncontact thermometer was 36.0 °C, the median was equal to 36.0 °C and standard deviation 0.6 °C.

After the normal distribution was found to match, an ANOVA analysis was performed. The *p*-value for the ANOVA means that among the compared measurement results, there are statistically significant different results. To show which measurements (which measurement method) are different from the others, a post hoc RIR Tukey test was performed. Analysis of the results of this test showed that there is a statistically significant difference between the results obtained in measurements with an infrared noncontact thermometer T4 and other methods of measurement T1, T2, T3. The temperatures measured with a noncontact thermometer were lower than the temperatures measured with other thermometers. Because temperature measurements with an infrared noncontact thermometer T4 took place in a different place of the subject’s body (on the forehead) than measurements made with contact thermometers T1, T2, T3 (armpit), intraclass correlation coefficient (ICC) analysis was performed. It is used in a situation where the measurements of the tested variable are made by several “judges”, in our case four different methods of measurement T1, T2, T3, T4. The ICC assesses the strength of reliability, that is, the degree to which their ratings are consistent. The obtained results indicate that there are no statically significant differences in the reliability of measurements. All used measurement methods had the same high reliability (Table 2).

The analysis of the correlation (linear Pearson) between the BMI index and the measured temperature was also performed. There was no correlation between the compared parameters.

The reliability of the compared methods is high, and the difference in temperature values measured with a noncontact thermometer in comparison with contact thermometers results only from a different place of measurement. Temperatures measured with contact thermometers in the lower armpit are higher than temperatures measured noncontact on the forehead. This does not mean hypothermia in patients with a lower temperature (measured on the forehead), but only indicates the correct distribution of temperatures on the surface of the body. In the armpit, with the arm closed, there are different thermal conditions than on the surface of the forehead, which is exposed.

The statistical analysis of the measurements shows that temperature measurements made with an infrared thermometer give statistically significantly lower values compared to temperature measurements made with other thermometers (mercury, mercury-free, electronic). The error of measurement methods should be considered a bit more broadly. It consists of a systematic error, largely dependent on the measuring device (which we mentioned in the introduction), and a random error, which is the standard deviation here. Small and comparable in all types of measurements made by us, the value of the random error indicates a well-planned and scrupulously conducted experiment. To facilitate the interpretation of the results, a Bland-Altman diagram was used, showing the degree of compatibility between two different samples—to estimate the 95% reproducibility coefficient. ‘Mean’ on the graph is the average of the differences. If the results obtained by two methods differ from each other, then there is a shift, called bias, and then the line showing the average of the differences is not at the level of 0, but is shifted significantly up or down from this level. The chart shows a 95% compatibility interval. If the differences are normally distributed, 95% of the differences will be in the range (mean of differences ± 1.96 SD), where SD is the standard deviation of the differences. Thus, the defined compliance interval is not the same as the confidence interval.

In Figure 1, the Bland–Altman chart indicates that an electronic thermometer gives higher results than an infrared thermometer on average by 0.2 °C (the line for the mean difference is 0.2 lower than the absolute correspondence illustrated by the 0 line). The range of the compliance interval is 2.3 °C.

In another analysis presented in the graph (Figure 2) by Bland–Altman, it was indicated that a mercury-free thermometer gives higher results than an infrared thermometer on average by 0.35 °C (the line for the mean difference is 0.35 lower than the absolute correspondence illustrated by the line of level 0). The range of the compliance interval is 2.57 °C.

Another Bland–Altman graph (Figure 3) for the collected data indicates that a mercury thermometer shows higher results (on average 0.22 °C) than an infrared thermometer (the line for the mean difference is 0.22 lower than the absolute correspondence illustrated by the line of level 0). The range of the compliance interval is 2.44 °C.

## 4. Discussion

The human body temperature values are distributed accordingly: constant temperature inside the body (deep temperature), variable temperature of the skin and extremities (surface temperature). In healthy people, a decrease in temperature values from the inside of the body to the shielding part is observed, because the temperature values of peripheral parts of the body depend on the conditions of the external environment and may vary from many degrees below to many degrees above zero [24]. From the surface of the body, heat is dissipated into the environment in several mechanisms: radiation, conduction, convection, evaporation. It is assumed that an organism in thermal comfort loses an average of 60% of heat by radiation, 15% as a result of heat conduction to the environment, 3% by conduction of heat from the body to other surrounding solids [25]. From the point of view of physiology, the human body is warm-blooded, thanks to which it is able to maintain a constant central temperature of about 37 °C regardless of external conditions. The center located in the hypothalamus is responsible for maintaining the right temperature, which supervises the production of heat in metabolic processes and its dissipation [26]. The thermoregulation center is responsible for maintaining a constant temperature of the internal organs, and the peripheral parts of the body change their temperature and can give away or retain heat depending on external conditions. In addition to temperature fluctuations in physiological ranges, changes in body surface temperature are observed primarily during the course of disease, including the course of COVID-19 infection [27]. Analyzing the somatic features of the studied group, attention was paid to the body mass index (BMI) in terms of the impact of this parameter on the values of body temperature measurements. Other researchers point to such a relationship [28,29]. The study also attempted to determine the relationship between the BMI value and the surface temperature of selected areas of the body depending on gender. There was no variation in body surface temperature between the studied group of women and men.

As mentioned, the distribution of the surface temperature of the body is not even, the distribution of values depends on the degree of blood supply to a given place of the body and the thickness of the subcutaneous fat tissue. This is important because both the skin, subcutaneous tissue, and in fact the subcutaneous fat layer form the thermoinsulator of the body. Other factors influencing the value of the measured temperature include gender, age, circadian rhythm, lifestyle, as well as individual predispositions of the body, e.g., rate and regulation of metabolism, emotional state [30].

Results of a study by Khorshid and others, assessing the accuracy of three types of thermometers [disposable (phase changeable), cylinder, mercury in glass], showed the consistency of the results obtained from their own tests, as to slight fluctuations in the values of measurements depending on the location of the test and the type of thermometer. Thus, higher readings measured with a mercury thermometer in glass were confirmed and the need to keep the thermometer in the armpit for a longer time (about 8 min) was determined [31].

The obtained temperature measurements made with an infrared thermometer gave statistically significantly lower readings compared to temperature measurements made with other thermometers (mercury, mercury-free, electronic). An electronic thermometer gives higher readings than an infrared thermometer by an average of 0.2 °C; a mercury-free thermometer gives higher readings than an infrared thermometer by an average of 0.35 °C, while a mercury thermometer shows higher readings than an infrared thermometer by an average of 0.22 °C (Figure 1, Figure 2 and Figure 3).

The differences we observed between the values of temperature measured with contact and noncontact thermometers were also observed by other scientists. Valle PC et al. measured the temperature with an infrared thermometer (in the ear) and investigated the correlation between this method and the measurement of rectal temperature with mercury and digital thermometers. They observed a significant difference in body temperature between measurements with an infrared thermometer and a rectal mercury thermometer. Based on their measurements, they concluded that in clinical conditions, an infrared thermometer has a very low sensitivity (accuracy) to detect fever, and in turn digital thermometry seems to be a good alternative to a mercury thermometer (for rectal measurements) [32].

Other scientists, Stave at al., analyzed the temperature measurements of 16 employees with noncontact infrared thermometers to identify COVID patients. Among the companies that tracked disease status, more than 15 million screening tests were performed, detecting only about 600 cases of fever. Of these, less than 53 cases of COVID-19 have been identified, accounting for 8% of people with fever. The authors conclude that temperature measurement is an ineffective tool and not recommended for COVID-19 detection [13].

In the case of checking a large number of people in public places, the first choice is noncontact thermometers that allow temperature measurement without any contact with the person. Analyzed research and systematic work presented by the Canadian Agency for Medicines and Health Technology indicates eardrum thermometers and thermal scanners, although attention has been paid to the accuracy of surface temperature measurements and the need for further studies [33]. The variation in average temperature values in individual regions of the body was analyzed, looking for a relationship with gender. Studies showed no significance of differentiation in BMI values, this indicator in the subjects was within the normal range and amounted to an average of 24.5. Other researchers point out that body temperature is strongly associated with markers of obesity in men and postmenopausal women. The lack of association in premenopausal women may be due to the influence of the menstrual cycle [34].

As shown, there was no correlation between BMI and temperature, which is probably due to the average BMI of the study group, and therefore, it does not seem to be related to changes in temperature values due to, for example, vascularation or muscle mass. Other researchers point to a correlation of BMI, gender, and age, but this is usually due to group selection and studies of older age groups. Matsumura and others indicate a correlation of lower BMI and muscle mass values in a group of people with a lower body temperature at baseline compared to normal body temperature. The mean age in the study group was 81 years. It was found that skeletal muscle mass, rather than metabolic rate or BMI, was strongly associated with an increase in body temperature after two years of exercise training [35].

We would like to emphasize that one of the limitations of thermometric testing in a large population may be due to the way it is carried out, i.e., the diligence of measurements, the lack of which may give results that differ significantly depending on the thermometric instrument used, and as a result, are burdened with error.

## 5. Conclusions

The methods used to measure body temperature with contact thermometers (mercury, mercury-free, electronic) and noncontact infrared thermometers statistically show high measurement reliability. The temperatures measured with contact thermometers in the upper armpit were higher than the temperatures measured noncontact on the forehead.

For the correct interpretation of the results, we should also pay attention to the fact that the measurements were made in different places. Temperatures measured with contact thermometers in the lower armpit are higher than temperatures measured with noncontact thermometers on the forehead. This does not signify hypothermia in patients with a lower temperature (measured on the forehead), but instead indicates the correct distribution of temperatures on the surface of the body. In the armpit, with the arm closed, there are different thermal conditions than on the surface of the forehead, which is exposed.

To correctly interpret the result of measuring human body temperature, it is necessary to specify the place of measurement and the type of thermometer used.

Due to the existing pandemic situation, it seems that telemedicine systems, used in some interdisciplinary fields of medicine, such as cardiovascular health informatics, may support the correct measurement of temperature, and thus the diagnosis of patients with suspected COVID-19. The inclusion of a panel with a correct temperature measurement method would be a simple and quick way to improve the diagnosis of COVID patients with comorbidities.

The authors plan to continue the research by extending the observations with the impact of the accuracy of temperature measurements in selected groups of patients with comorbidities, in particular in patients with cardiological and metabolic disorders.

## Figures and Tables

**Figure 1 healthcare-10-00331-f001:**
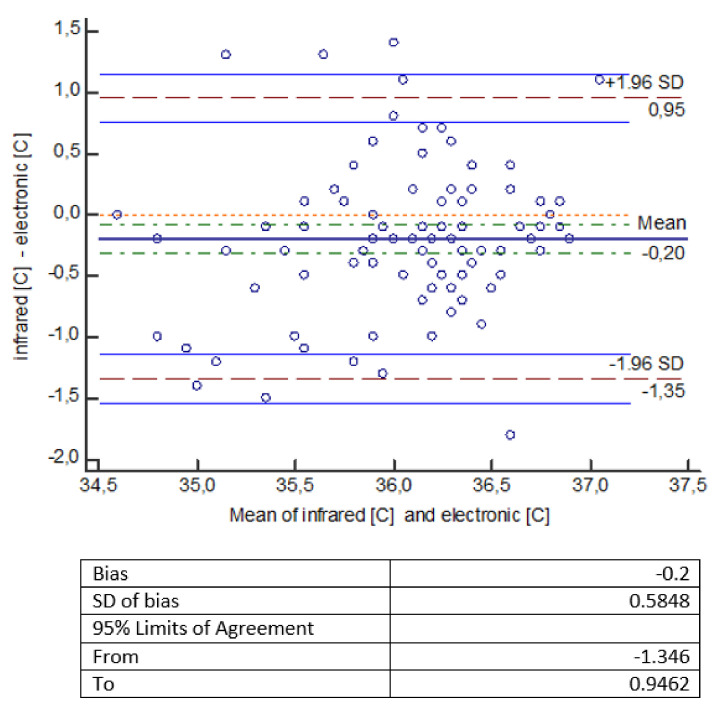
Bland–Altman diagram for temperature measurements [°C] made with thermometers: infrared and electronic.

**Figure 2 healthcare-10-00331-f002:**
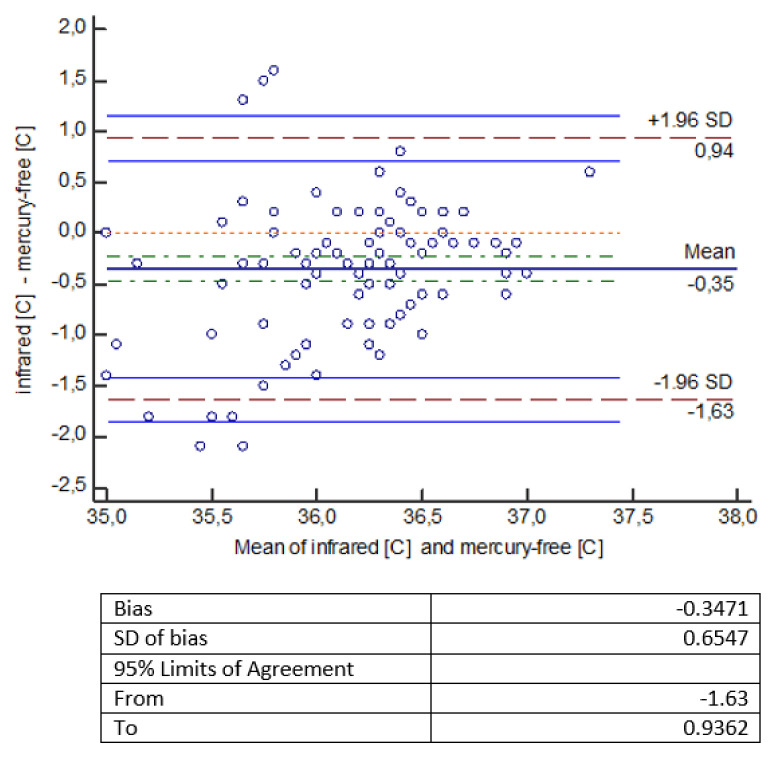
Bland–Altman diagram for temperature measurements [°C] made with thermometers: infrared and mercury-free.

**Figure 3 healthcare-10-00331-f003:**
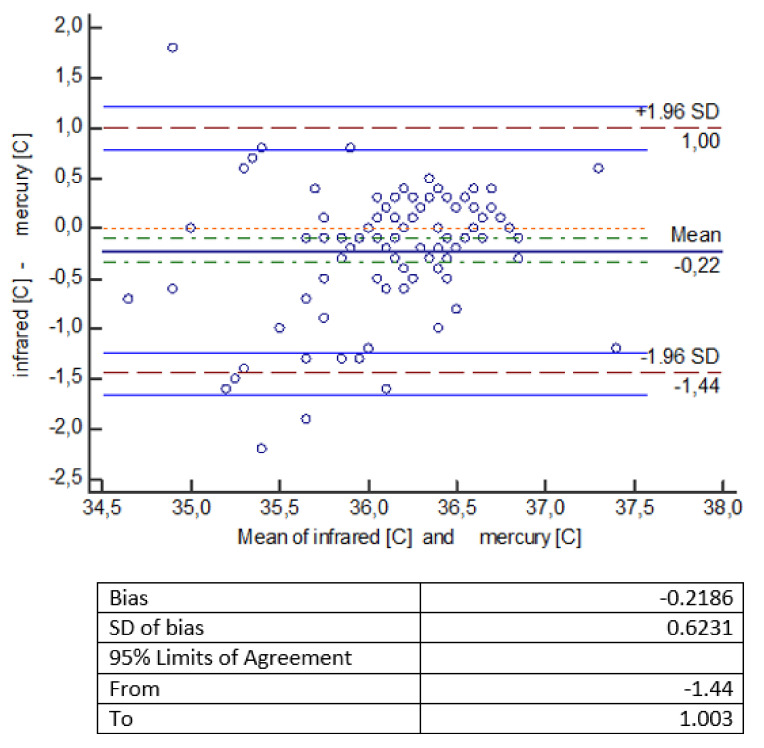
Bland–Altman diagram for temperature measurements [°C] made with infrared and mercury thermometers.

**Table 1 healthcare-10-00331-t001:** Characteristics of patients.

Variable	N	Average	SD	Median	Min	Max
**Age [years]**	102	36.0	14.5	38.0	38.0	38.0
**Height [m]**	102	1.72	0.09	1.72	1.72	1.72
**Body weight [kg]**	102	73.3	17.6	70.0	70.0	70.0
**BMI [kg/m^2^]**	102	24.5	4.6	23.6	23.6	23.6
**Gender**						
**F (%)**	50 (49)					
**M (%)**	52 (51)					

Abbreviations: SD—standard deviation.

**Table 2 healthcare-10-00331-t002:** Temperature values measured with various thermometers.

	Contact Thermometers [°C]			Noncontact Thermometers [°C]	P2 Value
Variable	T1 Mercury Thermometer	T2 Mercury-Free Thermometer	T3 Electronic Thermometer	T4 IR Thermometer	
**Average**	36.2	36.3	36.2	36.0	T1 vs. T2 *p* = 0.091 T1 vs. T3 *p* = 0.806 T1 vs. T4 *p* = 0.004
**SD**	0.5	0.5	0.5	0.6	
**Median**	36.2	36.4	36.3	36.0	
**Range**	34.0–38.0	35.0–37.2	34.5–37.5	34.4–37.6	T2 vs. T3 *p* = 0.053 T2 vs. T4 *p* < 0001 T3 vs. T4 *p* = 0.009
**P1 value**	*p* = 0.001				
**P3** **ICC** **value**	*p* (T1, T2, T3, T4) ≤ 0.001 ICC = 0.708 −95% CI for ICC +95% CI for ICC				

P1—ANOVA, test. P2—post hoc RIR Tukey test, P3—ICC test.

## Data Availability

The data presented in this study are available on request from the corresponding author.

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
