# Peer review of "Comparative Analysis of Human Body Temperatures Measured with Noncontact and Contact Thermometers"

_healthcare, 2022, doi:10.3390/healthcare10020331_

Round 1

Reviewer 1 Report

This study discusses the issue of differences in human body temperature measured by contact vs, non-contact thermometers and whether using of each type affect the accuracy and repeatability of measurements. This sounds important especially having a relatively wide range of measurement error between the two approaches that could have health consequences, particularly during the COVID-19 pandemic.     

Introduction

  • It is well-written, yet there is a lack in references (e.g. line 29) as many sentences were without references. I suggest authors to revisit it and cite additional references.
  • line 67 and 68: the sentence was repeated twice.
  • Authors highlighted the issue with using non-contact thermometers. They need to emphasize on the impact of external environmental factors such as weather or even sunlight on the body temperature and highlight how this make them less preferable compared to the contact thermometers.

Method:

Where was the study conducted? Was it reviewed by an IRB committee? What is the IRB #? Have participants been consented prior to starting the study?

Results:

  • Table 1: “N important”. Remove “important”
  • Line 182: one-hundred and two adults…. Instead 102 adults.
  • Line 182: all temperatures were normal, correct, and did not indicate fever or hypothermia instead of “The temperatures obtained in the measurements in all subjects were normal, correct, they did not indicate fever or hypothermia.”
  • Table 2: What is the reason of having temperatures based on the gender? Kindly change it to match the text and be clearer to the reader when you compare the results of measurement methods using ANOVA.
  • Line 197: it is known that p = 0.0015 indicates significance. Kindly remove it.
  • Table 3: make sure to have the correct format of numbers. For example p = 0,091 should be 0.091
  • Figure 1, 2 and 3: make sure to have the correct format of numbers. For example, mean - 0,20 should be - 0.20
  • Line 279 to 286: Irrelevant to the paper’s focus. You just need to mention if there is any correlation between BMI and body temperature with supporting evidence.
  • Line 297: 25,000 people instead of 25.000 people
  • Line 302: “of the mean values” instead of from the mean value
  • Line 304: change respectively: to respectively;

Discussion:

Authors did a fair job in comparing their findings with the literature.  

Author Response

Thank you for your review, the text has been corrected considering the suggestions made.

In response to comments:

  1. The authors created a questionnaire for this study, based on interview with the patient during a routine visit, after receiving a consent to participate in this survey. The study was performed anonymously, without collection patients’ personal data. The study was conducted as a survey that did not fulfill the medical experiment criteria therefore did not require Bioethical Committee approval.
  2. The format of the numbers in the graph was obtained automatically by the statistical calculation program. We couldn't change or fix the format. Therefore, under each chart, we used a table with the values entered with the correct format.

Reviewer 2 Report

The authors compare body temperature measurements with different sensor types and measurement sites at the body surface. The topic is of general importance in clinical use and of particular relevance for the COVID-19 pandemic. However, the manuscript has several flaws in methodology and the organization of the paper. 

Novelty

Measurement of body temperature has been investigated for several decades and received particular interest in the last month. These publications have not been covered as state of the art in the introduction. The innovation provided by the manuscript compared to the state of research is unclear.

Concept

The authors want to compare the reliability of different methods of body temperature measurement, with COVID-19 monitoring as main application. However, they only compare it for healthy patients: Most of the measurements are within a range of 1.5°C, only two samples exceeding 37.0°C. The confidence intervals in the Bland-Altman plots, in contrast, are larger than 2°C. So, you could obtain similar results with a thermometer that displays a fixed temperature value of, say, 36.5°C. In particular, the results do not tell anything about the ability of the thermometers to identify increased temperature.

The comparison between men and women are not valid: The authors report a large difference of mean age between the two groups (13 years, which is larger than the standard deviation of each group); for the other patient characteristics, no information is given concerning the two groups. Hence, the groups are not comparable.

The authors should also be aware that the measurement methods that they compare involve two determinants: the measurement site (forehead vs. armpit) and the sensor principle (infrared vs. mercury vs. ...). With the proposed measurements, these factors cannot be separated. However, the authors claim several times (e.g., l. 203 f., 225 f.) that the difference between the measurement methods are due to the sensors.

The authors also claim in the discussion that no correlation between BMI and temperature was found. However, they do not provide any information on that in the methods and results sections.

Finally, the authors claim in the discussion (l. 301 ff) to have found "lower values in women (...) than in men". That contradicts their findings in, e.g., Tab. 2, where no statistically significant differences are reported.

Organization of the manuscript

The manuscript shows the standard structure, and the sections "Materials and Methods" and "Results" are well organized. The Introduction and the Discussion, however, fail to address some of the most important points.

The introduction is intended to give a motivation for the research topic and an overview over the main concepts (both are provided in the manuscript) as well as a comprehensive and critical overview over the state of reseach, a description of the open question that the paper addresses and the innovation that is proposed. The latter points are completely missing in the introduction. Previous publications on the topic, particularly comparisons of different measurement site and/or different sensors are neglected. Furthermore, no information about the new insight that the paper shall provide is given.

The discussion section is intended to give a critical evaluation / discussion of the results, including limitations and interpretation of the results, as well as a comparison to related publications. The authors, in contrast, provide a full page with information about history, some previous publications and some fundamentals that should at most be included in the introduction. Only then a summary of the own findings and a comparison of some (not necessarily the most relevant) findings from other researchers is given.

Structure of arguments

Apart from the missing state of research and proposed innovation, the structure of the introduction is completely disorganized. It starts with the use of temperature measurement in clinical practice, goes to temperature scales, human skin temperature, sensor types, measurement duration, accuracy, again measurement duration, errors, temperature measurement in clinical practice again, and finally the question why temperature measurement should be of interest at all. The authors should consider the line of arguments before writing and delete any information that is not relevant (including repeated sentences).

Example: 10 lines of text (27-37) on the fact that there are different temperature scales with different offset and different numerical scales (or "steps") - an information that is not used in the rest of the manuscript. By the way: You cannot describe the difference between Kelvin and Rankine scale, when you only look at the zero point.

Methods

The "Materials and Methods" part comprises a lot of redundant information, e.g., on pressing of buttons etc. Relevant information, however, is missing: No information about the used devices is given. T3 is only specified as "electronic contact thermometer", without giving the sensor principle. Furthermore, it seems that there have been several measurements per patient, but without any information on that except for the 5min break between successive measurements.

Tab. 1: What should be the "important" in "N important"?

The description of measurement methods could be summarized in a table.

You should explain why you use mercury-based temperature measurement, when this method is withdrawn from production and sale for the last twelve years.

Results

Lines 185-193 are redundant, as they only repeat what is displayed in Tab. 2.

Table captions are positioned above (!) the table, not on the previous page.

The ANOVA is not necessary when a pair-wise comparison is done anyway.

Further remarks

  • Human skin temperature is also subject to heat flux from the environment to the body. That may be included in the term "heat released to the environment" but does not really sound like that.
  • The proposed description of the "sensitivity of the thermometer" does only make sense in the case of mercury-based thermometers.
  • Lines 86 ff.: The "correct" temperature in the armpit of a person with fever can be 39.1°C. "correct" is the wrong term here.
  • "Most often, non-contact thermometers are used, commonly referred to as non-contact."
  • The unit of temperature is ° (degree), hence you should use a proper degree symbol, not a superscript o (or sometimes not even superscript).
  • The test is called "Tukey" after John W. Tukey and does not have anything to do with a turkey.
  • The decimal separator in English language documents is a dot, not a comma.
  • Check the difference between a hyphen and a dash.
  • Variables and parameters, e.g. error probabilities, are set in italic font.
  • Lines 282 f.: The BMI is not the "ratio of body weight to body height"; it has been correctly described in the lines before.

References

  • Properly cite ref. [1] in the list of references. It is a chapter from a book, hence you should cite the book as a book.
  • Ref. [18]: Write out CDC.
  • Ref. [27]: Remove space before Bastardot.

Author Response

Thank you for your review, the text has been corrected considering the suggestions made.

In response to comments:

  1. The authors want to compare the reliability of different methods of body temperature measurement, with COVID-19 monitoring as main application. However, they only compare it for healthy patients: Most of the measurements are within a range of 1.5°C, only two samples exceeding 37.0°C. The confidence intervals in the Bland-Altman plots, in contrast, are larger than 2°C. So, you could obtain similar results with a thermometer that displays a fixed temperature value of, say, 36.5°C. In particular, the results do not tell anything about the ability of the thermometers to identify increased temperature.

Ad:1 The aim was to compare the surface temperatures of the human body measured by various techniques, in particular a non-contact thermometer (infrared) and contact thermometers (mercury, mercury-free, electronic). A novelty in our research was that the temperature was measured under controlled conditions using several methods (thermometers) in the same healthy people. The assumption of our work was not to compare temperatures in sick patients (with fever from various causes or with COVID-19), but only to indicate the reliability of the measurements. The possibility of using the selected measurement technique, for example in monitoring COVID-19, is left to the readers. "So, you could obtain similar results with a thermometer that displays a fixed temperature value of, say, 36.5 ° C" - I don't think I understand that.

  1. The comparison between men and women are not valid: The authors report a large difference of mean age between the two groups (13 years, which is larger than the standard deviation of each group); for the other patient characteristics, no information is given concerning the two groups. Hence, the groups are not comparable.

Ad:2 As suggested by the reviewer, we do not compare the temperature of men and women separately.

  1. The authors should also be aware that the measurement methods that they compare involve two determinants: the measurement site (forehead vs. armpit) and the sensor principle (infrared vs. mercury vs. ...). With the proposed measurements, these factors cannot be separated. However, the authors claim several times (e.g., l. 203 f., 225 f.) that the difference between the measurement methods are due to the sensors.

Ad:3 We included some explanation in the text, but perhaps it was not emphasized enough. In line with the reviewer's comments, we will make it clear (this time in the conclusion section).

  1. The ANOVA is not necessary when a pair-wise comparison is done anyway.

Ad:4 The ANOVA test was used to compare repeated measures to see if there were differences within the group. After detecting the differences, the Tuckey test checked between which thermometers there were differences. You cannot use pairs comparison alone because the error of the comparison would add up.

Reviewer 3 Report

  • The author has mentioned the errors obtained by used techniques, it is suggested that the significance of errors listed, must be described in the comparison section.
  • The major trends of the simulation results may be shown in the bullet form.
  • Comparsion with recent study and methods would be appreciated. 
  • Introduction section can be extended to add the issues in the context of the existing work and how proposed algorithms/approach can be used to overcome the problem.
  • The paper does not provide significant experimental details needed to correctly assess its contribution: What is the validation procedure used?
  • Quality of figures is so important too. Please provide some high-resolution figures. Some figures have a poor resolution.
  • More motivation/context regarding the application side of it, particularly on the aspects that make this technique particularly suited for industrial application scenarios, and how it would be applied in real scenarios. These aspects could additionally be supported with some related work context. Such as Challenges and opportunities in cardiovascular health informatics.
  • Conclusion should state scope for future work.
  • Results need more explanations. Additional analysis is required at each experiment to show the its main purpose.

Author Response

Thank you for your review, the text has been corrected considering the suggestions made.

In response to comments:

The paper does not provide significant experimental details needed to correctly assess its contribution: What is the validation procedure used? 

Due to the exacerbated sanitary situation and changing epidemic conditions related to the COVID-19 pandemic, we were unable to carry out repeat temperature measurements on the same group of volunteers. Please indicate which ditals we should put - maybe they are in our database, and we will be able to enter them.

Round 2

Reviewer 3 Report

none

Author Response

Dear ‘Reviewer 3’,

Thank you for your positive review.